# Strain Measurement in Single Crystals by 4D-ED

**DOI:** 10.3390/nano13061007

**Published:** 2023-03-10

**Authors:** János L. Lábár, Béla Pécz, Aiken van Waveren, Géraldine Hallais, Léonard Desvignes, Francesca Chiodi

**Affiliations:** 1Thin Film Physics Laboratory, Institute of Technical Physics and Materials Science, Centre of Energy Research, Konkoly Thege M. u. 29-33, H-1121 Budapest, Hungary; pecz@mfa.kfki.hu; 2Centre de Nanosciences et de Nanotechnologies—C2N, Université Paris-Saclay, CNRS, 91120 Palaiseau, Francefrancesca.chiodi@c2n.upsaclay.fr (F.C.)

**Keywords:** STEM, reciprocal space lattice fitting, free computer program, superconducting silicon, strain components, concentration of substitutional atom, boron in silicon

## Abstract

A new method is presented to measure strain over a large area of a single crystal. The 4D-ED data are collected by recording a 2D diffraction pattern at each position in the 2D area of the TEM lamella scanned by the electron beam of STEM. Data processing is completed with a new computer program (available free of charge) that runs under the Windows operating system. Previously published similar methods are either commercial or need special hardware (electron holography) or are based on HRTEM, which involves limitations with respect to the size of the field of view. All these limitations are overcome by our approach. The presence of defects results in small local changes in orientation that change the subset of experimentally available diffraction spots in the individual patterns. Our method is based on a new principle, namely fitting a lattice to (a subset of) measured diffraction spots to improve the precision of the measurement. Although a spot to be measured may be missing in some of the patterns even the missing spot can be precisely measured by the lattice determined from the available spots. Application is exemplified by heavily boron-doped silicon with intended usage as a low-temperature superconductor in qubits.

## 1. Introduction

Strain alters the physical properties of single crystals. This fact has long been utilized in semiconductor devices, where, e.g., mobility is enhanced in strained Si-Ge devices [1]. Other applications of strain in microelectronics are mentioned in a recent review of the measurement of strain by high-resolution X-ray diffraction (HRXRD) [2]. Electron microscopes also play a significant role in the determination of strain in single crystal parts (e.g., layers) in devices. Although the spectral resolution of electron diffraction in the transmission electron microscope (TEM) is inferior to that of HRXRD, it has two advantages. First, it gives local information. Second, the area for local measurement can be selected from an image, which shows the local real space structure of the sample. Hytch and coworkers developed a method to measure strain from high-resolution transmission electron microscope (HRTEM) images [3]. The method that is based on fast Fourier transformation (FFT) of the HRTEM image later became known as geometric phase analysis (GPA) [4]. A similar method, using real space processing of the HRTEM image is called the peak pairs algorithm [5]. All HRTEM-based methods can only process a limited field of view (FOV) inherent in the number of pixels in the camera and the simultaneous requirement that the recorded image must resolve the crystal lattice. Balboni and coworkers used convergent beam electron diffraction (CBED) in the TEM to measure strain in heterostructures [6]. However, in that form of processing the measurement is individual in manually selected points and does not lend itself to mapping. Both Beche and coworkers and Hytch and coworkers applied dark field electron holography for strain measurement [7,8]. Although it can provide the distribution of strain over a large area, it requires hardware (holography), which is less frequent in TEMs. Pekin and coworkers mapped strain from scanning nanobeam diffraction and used a disk registration algorithm to determine the accurate positions of the individual disks (that have finite disk size due to the convergence of the used electron beam) [9]. Commercial solutions are also offered for the mapping of strain [10,11]. Although several methods are available, there remained a need for a method that overcomes the above-mentioned limitations. We report such a solution here.

A new computer program was developed to determine strain components and concentration of the substitutional dopant from 2D diffraction patterns collected in a STEM from a 2D area of a single crystal sample (TEM lamella), constituting a 4D-ED dataset. The dataset contains *n×m* diffraction patterns, where *n* is the number of columns and *m* is the number of lines in the STEM image. The important point is the spatial distance between the locations where the individual diffraction patterns are recorded. By selecting the step size of the STEM (between neighboring pixels) in accordance with the probe size, we can maximize information content. In that way, both the spatial resolution of the STEM image and the spatial resolution of the strain map are determined by this step size that must be in accordance with the need, which spatial resolution strain map is adequate for the given sample. The number of columns and lines is determined by the size of the sample that needs to be mapped. The possibilities and limitations of this program are elaborated in the paper exemplified with application to silicon doped with boron well above the solubility level. The significance of such examination is that the development of a new Qubit is underway within a European Project (SIQUOS). The aim is to develop a scalable technology, which is fully compatible with existing Si-technology platforms, materials, and processes. The targeted Qubit will consist of a MOSFET with a superconducting source and drain. One of the superconducting materials in the study is Si:B, where Si is doped much above the solubility level by methods summarized in the paper. The new program is distributed free of charge from web site https://public.ek-cer.hu/~labar/Strain_4D-ED.htm (accessed on 1 March 2023). Its new lattice fitting approach even facilitates measurement of the position of diffraction spots, which may not be excited in all of the diffraction patterns of the 4D-ED dataset. Its advantages over other existing methods are that it does not require special hardware (in contrast to dark field holography), can measure larger areas (than HRTEM-based methods), and is freely distributed (in contrast to commercial methods).

Section 2 outlines what kind of instrumentation and samples are used when the operation of the program is exemplified. Section 3 elaborates on the principles of the method, the typical workflow, and the operation of the program. Section 4 gives an example of results obtained from Si:B samples.

## 2. Instruments and Materials

TEM lamellae were prepared with a ThermoFisher Scios 2 Dual Beam microscope (Eindhoven, The Netherlands) with EasyLift^TM^ nanomanipulator.

Experimental data, input to the Strain4DED program were collected in a Titan Themis G2 200 (Thermo Fisher Scientific, Waltham, MA, USA) STEM equipped with X-FEG gun and 4 k × 4 k CETA 16 CMOS camera (Thermo Fisher Scientific, Waltham, MA, USA). The collection of the diffraction patterns for the 4D-ED dataset was controlled in “microprobe STEM mode” by TIA software (FEI, Eindhoven, The Netherlands). This mode provides a nanometer-sized, almost parallel electron beam (convergence angle 0.24 mrad), scanned over a selected part of the TEM lamella. Although the spatial resolution of the STEM high-angle annular dark-field (HAADF) image is almost an order of magnitude poorer in this mode than in usual STEM, the appearance of the diffraction patterns (that contain small spots in contrast to extended discs) is similar to that of selected area electron diffraction (SAED) patterns. This fact means that the strain maps produced by the program can have ~1 nm spatial resolution. TEM images were recorded with the same CETA camera as the diffraction patterns. STEM Z-contrast images were recorded with a HAADF detector (E.A. Fischione Instruments, Inc., Pennsylvania, PA, USA) controlled by Velox software (version 3.3, FEI, Eindhoven, The Netherlands). The usual dwell time while the beam is in a position that corresponds to a pixel of the HAADF image is 100 ms. Spot size 8 results in strong enough peaks in the diffraction pattern to make processing unambiguous. Even though the peaks may be saturated, their intensity is below the level that could damage the camera. Application of a hybrid pixel detector in other instruments facilitates recording even higher intensities. Pixel size (controlled by binning) and camera length must be selected in a way that the first 2 to 4 orders of diffraction spots extend over the entire camera. We used only a quarter of the camera with binning 4 (resulting in 256 × 256 pixels for a diffraction pattern). Calibrated camera length appropriate for measurement of Si close to the [110] zone orientation proved to be 262 mm. The size of the area for the 4D-ED experiment is determined by selecting a part of a larger image simultaneously with selecting the number of columns and rows for the STEM. They also yield the size and the spatial resolution of the HAADF image (and the corresponding strain map at the end). A strain map with 200 × 100 pixels (resulting in 20,000 diffraction patterns to process) seems to be satisfactory. It corresponds to about 200 nm × 100 nm mapped area with our spatial resolution.

The computer code for the Strain4DED program was IId in MS Visual Basic and runs under the Windows operating system on PCs. The self-contained executable program can be downloaded free of charge from https://public.ek-cer.hu/~labar/Strain_4D-ED.htm (accessed on 1 March 2023). Instructions for installation are also provided on the same home page. Either the program or the home page does not collect any data from the visitors or users, maximally satisfying GDPR requirements.

The sample used here as an example of operation of the program originates from the SIQUOS project (Quant-ERA program Grant Agreement No 101017733 of the EU). The sample in this paper is monocrystalline Si:B layers epitaxied on bulk (100) Si. In order to induce superconductivity, extreme doping levels are needed, more than three times the solubility limit (n_solubility_~4 × 10^20^ cm^−3^), so that a non-equilibrium technique, such as nanosecond laser doping, is required. We employ gas immersion laser doping (for a review see [12]): a precursor gas (BCl_3_) is injected into an ultra-high vacuum chamber, where it saturates the chemisorption sites on the Si surface. The silicon is melted up to the desired doping depth by a XeCl 308 nm excimer laser of 25 ns pulse duration and tuneable energy. The melting of Si induces the rapid diffusion of the chemisorbed B into the liquid Si region. Due to the high recrystallization velocity (4 m/s [13]), the Si undergoes quenching, and high dopant concentrations beyond the solubility limit can be reached. As a result, a homogeneously doped box-like Si:B crystal is grown by liquid phase epitaxy on the underlying Si substrate. Multiple chemisorption/melting cycles determine the number of dopants introduced in the doped layer [14]. In this work, an Si:B layer of total thickness d = 121 nm is investigated. The thickness is determined by cleaving the doped spot in the middle and performing a KOH chemical etch at 80 °C for 45 s on the cleaved edge. The KOH etches away the underneath Si substrate, while the p-type ultra-doped Si:B acts as an etch stop. The hanging ledge is then observed in a scanning electron microscope at 90° angle to determine the thickness of the highlighted Si:B layer. The found thickness is confirmed within a 10% error by the strain analysis, as a deformation in the normal direction, induced by the B substitutional incorporation, is observed over the top 127 nm. The energy surface density necessary to melt d = 121 nm is E = 960 mJ/cm^2^. This energy is determined from the laser energy temporal profile, measured in an integrating sphere with a nanosecond photodiode. The energy is calibrated through the comparison, in situ, with the reference melting energy of an Si monocrystalline (100) undoped substrate placed near the doped samples [15]. The described doping process is repeated 300 times, in order to reach an active concentration n_B_ = 3.06 × 10^21^ cm^−3^ (6.1 at.%). The active concentration is determined by Hall measurements, with a Hall factor of 0.75 [16]. The total B concentration C_B_ (which includes both electrically active—i.e., substitutional—and inactive B atoms), C_B_ = 4 × 10^21^ cm^−3^, is known from a series of secondary ion mass spectroscopy (SIMS) measurements on Si:B samples with a varied number of doping cycles (1 to 700) and thickness (65 nm to 176 nm). As expected, C_B_ increases linearly with the number of doping cycles: indeed, the amount of B introduced at each cycle is determined by the constant number of chemisorption sites, which are saturated before the laser melting. The activation ratio (n_B_/C_B_), is in this sample 77%, highlighting the large activation achievable by nanosecond laser doping, even at the extreme doping concentrations performed. The superconducting critical temperature of the sample is T_c_ = 0.35 K, determined by low-temperature resistance vs. temperature measurements in an adiabatic demagnetization refrigerator. The superconducting critical temperature depends on both the active B concentration and the deformation of the Si:B layer [17]. Thus, a precise determination of the deformation profile in the layer thickness is essential for a correct understanding of the establishment of superconductivity in superconducting silicon.

The layer contains many crystal defects (dislocation, stacking faults (SF), and nano-twins) as can be seen in Figure 1a. The characteristic distortion caused by the substitutional B in the Si lattice is different in directions normal to the layer and parallel to it (lateral). The concentration of the substitutional B content is determined from these two strain components using Poisson’s ratio for Si. Detailed evaluation of the development of strain and B concentration as a function of processing is a topic of a separate publication.

## 3. The Strain4DED Method

### 3.1. Basic Principles and Workflow

The method harmonizes spatial and spectral resolution in the diffraction experiment and determines strain components from the shift in the position of diffraction spots in locally recorded spot diffraction patterns. A nanometer sized almost parallel electron beam (convergence angle 0.24 mrad) produces diffraction spots (in contrast to disks in nanobeam electron diffraction, which are usually recorded with larger convergence to improve spatial resolution). This setup results in a compromise between spatial resolution (limited to 1 nm) and spectral resolution, which is improved by the presence of well-defined spots. Correct localization of these spots is further improved by fitting a full lattice to the subset formed by all detected spots in a measured 2D pattern. (The significance of that approach is elaborated in relation to Figure 2a.) The achieved sub-pixel precision ensures the required precise measurement of strain (relative shift in position of lattice points). Since the measurement is relative (to that measured in an unstrained region) any small distortion of the diffraction pattern (that might be caused by the lenses of the TEM) can only cause second-order effect in measured strain. Even this second-order error can be corrected for with one of the methods published by Lábár [18,19,20]).

In our usual workflow, first, we record a TEM image (Figure 1a) to show an area with good spatial resolution. A bright field (BF) image gives visual enhancement of defects (Figure 1b). Next switch to “microprobe STEM mode” (select the beam current not to destroy the camera but provide enough count in the diffraction spots, mark the area for data collection) and collect the 4D-ED dataset. We record a poor resolution HAADF image with this nearly parallel beam (Figure 1c) simultaneously recording the 4D-ED dataset. This shows accurately which area was used for the data collection. Our usual experimental parameters produce strong enough peaks in the diffraction pattern (Figure 1d). Even the poorer spatial resolution of the HAADF image is enough to identify the area in the previously recorded good resolution images. For FIB-lamellae the presence of the Pt protecting layer on top of the sample further helps exact identification of the location of the examined area.

In CMOS cameras (like ours) the pixel size is controlled by binning. The binning in CMOS cameras is software driven (in contrast to the hardware solutions in CCDs). CMOS cameras can be indirectly coupled (via optical fiber or mirror) or direct electron detection. Although their dynamic range (maximum counts in a pixel) is more limited than in hybrid pixel cameras they are also suitable for 4D-ED experiments for many sample types. Camera length must be selected together with pixel size (and the number of pixels used in the camera) to achieve that minimum the first two orders of spots in the pattern are recorded. There is no need for more than 4 orders of spots since the larger angular range simultaneously also means a reduction in the spectral resolution (by reducing the distance of spots expressed in number of pixels). The first 3 orders are clearly visible in Figure 1d (and there is room for the unexcited 4th order within the useful area of the camera).

The large number (e.g., 20,000 for a 200 × 100-pixel image) of measured diffraction patterns are saved in uncompressed Tif format for serving input to the Strain_4D-ED program. Processing is separated to two distinct steps. First, a peak search algorithm is carried out on all patterns. The second step of lattice fitting is based on the saved list of found peak positions. Strain components (ε_xx_, ε_xy_, ε_yx_, ε_yy_ and the derived quantities, shear and rotation) are determined from the positions of the lattice spots, identically to the procedure used in different variants of the HRTEM-FFT-based GPA method [3,21].

### 3.2. Details of Operation

#### 3.2.1. Peak Finding Algorithm

The list of peaks is derived in two steps. First, all pixels are identified, which contain more counts than the background (BKG) by a pre-defined multiples of the standard deviation of BKG. The BKG is estimated around any square-shaped area as the average counts in the perimeter of the square. Each pixels exceeding the threshold are registered first. The square is stepped with smaller amount than the size of the square to produce overlap. In the next step, the registered pixels are grouped. All registered pixels, which are each other’s neighbors are grouped to form a single peak. Redundantly listed pixels (if any) are omitted during that step. The weight of mass of the component pixels gives the coordinate of the peak center. Groups formed by less pixels than a pre-defined limit are disregarded. Finally, the found peaks are ordered by their distance measured from the central beam (i.e., by d-value of the corresponding Bragg reflection). The default parameters of the peak finding algorithm (multiplier of standard deviation, minimum number of pixels in a peak) can be manually edited or the default values restored. The peak finding procedure (which is the longest part of the procedure) takes about 10 min for 20,000 patterns in an 8-year-old laptop (Intel(R) Core (TM) i5-3337U CPU @ 1.80 GHz). It means the number of patterns can be further increased if needed, especially with faster computers. Our program can also visualize the identified spots (drawn as circles) overlain the measured pattern.

#### 3.2.2. Fitting of the Lattice

The lattice is defined by 3 spots, the central beam and the endpoints of the 2 shortest reciprocal lattice vectors. Numbers 0, 1, and 2 in Figure 2a mark such three points. All spots of the lattice are formed by endpoints of linear combinations of these two vectors (01 and 02 base vectors) with integer coefficients. Usually, the endpoints of the base vectors are different from the spots, whose positions define the individual strain components (in our case laying horizontally and vertically from spot of the central beam, 0 marked as N and L in Figure 2b). You can see that the spot corresponding to normal direction (N) is only poorly visible, especially in Figure 2a, however its positions is well defined by the lattice. The lattice is built by arrays of two integer numbers (n1(spot number) and n2(spot number)), where spot number is a sequence number of registered lattice points. Although the entire lattice is defined by all integer pairs −4 ≤ n1 ≤ 4 and −4 ≤ n2 ≤ 4, fitting is only performed for those, which correspond to well-detectable (excited) diffraction spots in the given pattern. Due to the occasional bending of the TEM lamella, combined with the effect of crystal defects (e.g., dislocations) the local orientation of the sample slightly changes from pixel to pixel. Even though we remain close to the originally selected crystal orientation (usually a zone axis orientation) the different diffraction spots may be excited differently in the individual patterns, as illustrated in Figure 2. In that way, the same full lattice is fitted to different subsets of spots in the subsequent patterns.

That procedure ensures that the position of lattice point is precisely defined and can be measured even for spots, which are not visible in a particular pattern (e.g., the almost horizontally positioned point, N defined by the endpoint of the sum of 01 + 02 vectors in Figure 2a).

Figure 3 presents the logic of fitting full lattice to the measured subset of spots.

#### 3.2.3. Calculating Strain Components

Strain components are directly calculated from the measured position of the diffraction spot (the length (*g*) of the reciprocal lattice vector, *g* pointing to the diffraction spot).
(1)$$\varepsilon =\frac{g_{ref}-g_{strained}}{g_{strained}}$$

We select the strain component by selecting the direction of the *g*-vector in (1). In most of the literature (which determines strain from TEM) the strain component normal to the layer (surface normal) is named *ε*_xx_ and the lateral component is called *ε*_yy_. In our case the *ε*_xx_ is calculated from the length of the *0N* vector and *ε*_yy_ is from the *0L* vector. Similarly, the cross partials *ε*_xy_ and *ε*_yx_ measure the movement of the diffraction spot perpendicular to the direction of the *g*-vector. If *u_x_* and *u_y_* represent the components of the shift vector of the spots (*g*_ref_-*g*_strained_),
(2)$$\varepsilon_{xx}=\frac{\partial u_{x}}{\partial x}\;\varepsilon_{yy}=\frac{\partial u_{y}}{\partial y}\;\varepsilon_{xy}=\frac{\partial u_{x}}{\partial y}\;\varepsilon_{yx}=\frac{\partial u_{y}}{\partial x}.$$

Shear and rotation are calculated from the cross partials.
(3)$$\mathrm{Shear}=\frac{1}{2}\left(\frac{\partial u_{x}}{\partial y}+\frac{\partial u_{y}}{\partial x}\right)\;\mathrm{Rotation}=\frac{1}{2}\left(\frac{\partial u_{y}}{\partial x}-\frac{\partial u_{x}}{\partial y}\right)$$

#### 3.2.4. Presentations of Strain Components

The program renders the 2D distribution of the selected strain component color-coded on the screen. Values outside of the selected range are white. Pixels where the lattice cannot be determined (such as an amorphous part or region of the nanocrystalline protective Pt coating, which is not part of the original sample) are displayed as black. A color key and a size marker are drawn beside the 2D strain distribution. Figure 4 shows example of this rendering. The images are also saved in BMP format.

Depth distribution of the strain components as a 1D function of depth is also calculated and saved. Pixels are averaged along lateral direction to obtain the depth distribution. An example of depth distribution of normal and lateral strains (calculated from data in Figure 4) is shown in Figure 5. The data are also saved in a text file for alternative presentation or any other usage.

#### 3.2.5. Concentration of Substitutional Element

Concentration calculation is based on the assumption that the distortion is caused by substitutional element (one element is dissolved in the other where both pure elements have the same crystal structure, in our case virtual cubic B, see below). In such binary solid solutions, Vegard’s law can be applied:(4)$$a_{solution}=c\cdot a_{solute}+\left(1-c\right)\cdot a_{matrix}$$
where a is the corresponding lattice constant and c is the atomic fraction of the solute element. The matrix is Si and *a_Si_
*= 0.5431 nm is taken from XRD database card #04-002-0118. Elementary B is not available in cubic form, however, an Si-1%B is found in XRD database NIMS_MatNavi card #4295421637_1_2 with *a_Si-1%B_
*= 0.54166 nm. From this value, a virtual pure B cubic crystal was extrapolated resulting in *a_solute_
*= *a_B_*= 0.40009 nm. It is between the extreme values for cubic B lattice values reported in the literature (from *a_B_* = 0.378 nm [22] to *a_B_* = 0.4084 nm [23]. In an isotropic crystal, the lattice constants deformed in normal and lateral directions are connected by Poisson’s ratio to the unstrained lattice constant:(5)$$\frac{a_{matrix,Normal}-a_{matrix}}{a_{matrix}}=K\cdot \frac{a_{matrix,Lateral}}{a_{matrix}}$$
where $K=\frac{-2\cdot \nu }{1-\nu }$ and ν is Poisson’s ratio.

When combined with Equation (2), we can express the atomic fraction of the solute from the two strain components as
(6)$$c=\frac{a_{matrix}}{\left(1-K\right)\cdot \left(a_{solute}-a_{matrix}\right)}\cdot \left(K\cdot \varepsilon_{matrix,Lateral}-\varepsilon_{matrix,Normal}\right)$$

Default values for Si as matrix and B as solute are incorporated into the program. Relevant values for other element pairs can be manually input at the start of the calculation. Depth distribution of the solute concentration is also stored in the same text-file. Figure 6 shows the depth distribution of concentration of boron in silicon matrix calculated from the same data set as the previous figures.

## 4. Discussion

As the concentration is calculated using both components of strain, its value, compared to concentration values from an independent, different physical measurement gives a good estimate of the reliability of that quantity.

The concentration of the electrically active B, averaged over the doped layer thickness d = 121 nm, is determined by Hall measurements to be 3.06 × 10^21^ cm^−3^. This is compared to the concentration of substitutional element (B) in Figure 6 since it is assumed that B at substitutional sites corresponds to the active B quantity. The agreement is good, in particular in the top 90 nm, where the two values are within 7 % relative. The discrepancies may be due to the choice of *a_B_*, or to the use of the Si Poisson coefficient, which may be modified in ultra-doped Si:B. However, such curves give important information, drawing attention to a fully strained (zero lateral deformation), lower doping, and 30 nm region at the bottom of the layer. Since superconductivity only appears in relaxed layers, such observation points out that superconductivity does not extend through the whole doped layer. In-depth characterization of Si:B samples from the point of view of superconductivity, electrical properties, and applications is the topic of separate papers.

Since the size of the sample selected for Figure 2, Figure 3, Figure 4, Figure 5 and Figure 6 facilitates HRTEM examinations as well, we also compared our measured strain values to those obtained from HRTEM images by the Strain++ method [21]. Since this method measures the same thing by an independent approach, it also provides a test of the reliability of our method. These results are presented in Figure 7. Comparison to Figure 6 demonstrates a good agreement.

## 5. Conclusions

The 4D-ED technique for strain measurement, which is introduced here, gives a reliable distribution of strain components. It is distributed free and does not need special instrumentation (such as holography). It does not restrict the size of the measured area within the TEM lamella. Reference areas may be measured separately. These features overcome the limitations posed by the HRTEM-based similar methods.

## Figures and Tables

**Figure 1 nanomaterials-13-01007-f001:**
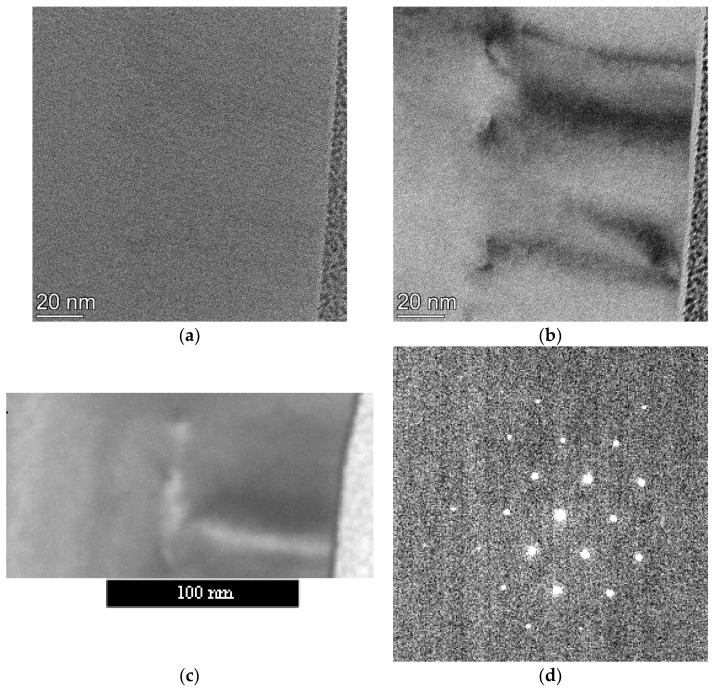
Example of selecting area and collecting 4D-ED data set: (**a**) HRTEM image of an area of the sample; (**b**) BF image of the same area with enhanced contrast around defects; (**c**) STEM image from the sub-area from where the 4D-ED dataset is collected; (**d**) Diffraction pattern recorded with the nearly parallel (α = 0.24 mrad) beam from an area of 1 nm diameter (the size of the beam in this mode). A total of 20,000 such patterns form the 4D-ED dataset.

**Figure 2 nanomaterials-13-01007-f002:**
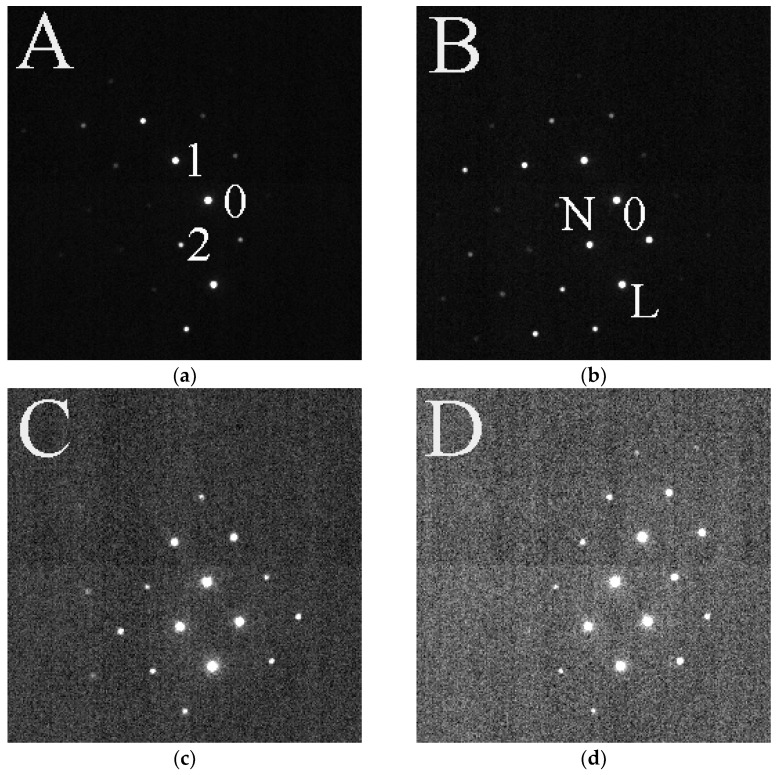
Illustration of how different diffraction patterns can be measured within a 400 nm wide region of a Si sample. Locations are marked on (**g**). (**a**) The pattern recorded at location A. Three diffraction spots are marked with numerals (0 is the central spot, 1 and 2 mark the endpoints of the two shortest reciprocal lattice vectors) (**b**) The pattern recorded at location B. Three diffraction spots are marked with characters (0 is the central spot, N and L mark normal and lateral directions from the central beam) (**c**) The pattern recorded at location C. (**d**) The pattern recorded at location D. (**e**) The pattern recorded at location T, which is a position of nanotwins. (**f**) The pattern recorded at location F, which is the position of C-Pt protecting layer deposited during FIB preparation. (**g**) STEM HAADF image of the 400 nm wide region. Letters mark the locations where the diffraction patterns were collected.

**Figure 3 nanomaterials-13-01007-f003:**
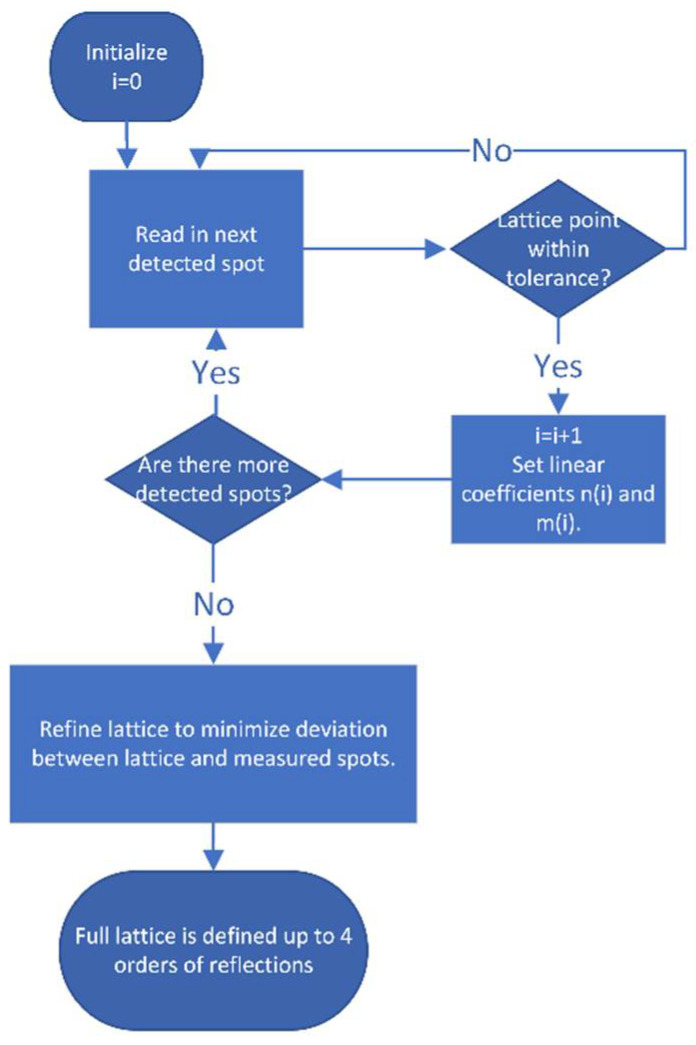
Block diagram showing the steps of fitting the full lattice to the measured subset of spots.

**Figure 4 nanomaterials-13-01007-f004:**
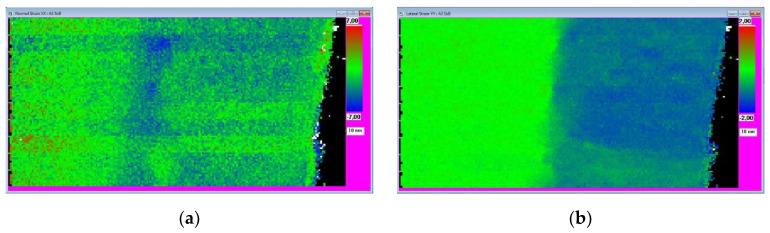
Two-dimensional strain distribution in an Si:B sample. The layer is on a bulk substrate. The top surface (covered with protecting Pt during FIB cutting) is on the right side of the map. The 4D-ED dataset was recorded from the area shown in Figure 1c. Pixels in the map are black, where it was impossible to fit a grid to the corresponding pattern (e.g., protecting layer). (**a**) Normal component. (**b**) Lateral component.

**Figure 5 nanomaterials-13-01007-f005:**
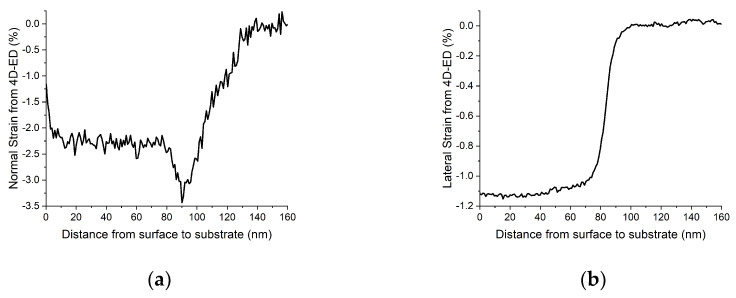
One-dimensional strain distribution in an Si:B sample as a function of depth. (**a**) Normal component. (**b**) Lateral component.

**Figure 6 nanomaterials-13-01007-f006:**
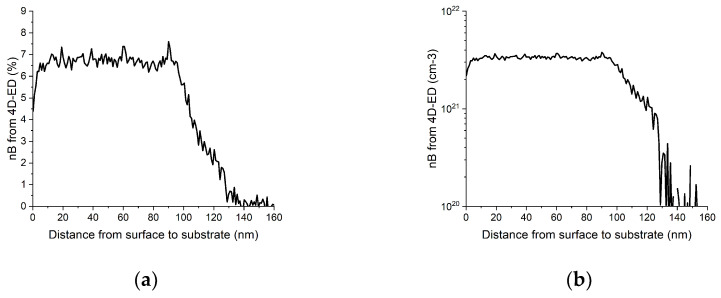
One-dimensional depth distribution of B in the Si:B sample. (**a**) Linear scale concentration (at%) to better show details. (**b**) Logarithmic scale (atom/cm^3^) more usual presentation in semiconductor physics.

**Figure 7 nanomaterials-13-01007-f007:**
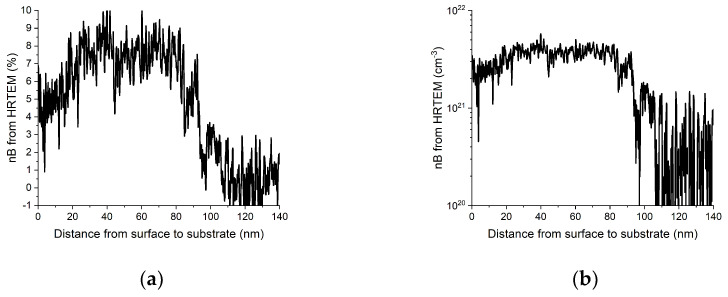
One-dimensional depth distribution of B in the same Si:B sample (as in Figure 6) measured from HRTEM images by the Strain++ method [21]. (**a**) Linear scale concentration (at%) to better show details. (**b**) Logarithmic scale (atom/cm^3^) more usual presentation in semiconductor physics.

## Data Availability

The reported program distributed free of charge from the home page of the first author. https://public.ek-cer.hu/~labar/Strain_4D-ED.htm (accessed on 1 March 2023).

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
