# Peer review of "Strain Measurement in Single Crystals by 4D-ED"

_nanomaterials, 2023, doi:10.3390/nano13061007_

Round 1
Reviewer 1 Report
In their paper "Strain measurement ...", Janos L. Labar and coworkers report on a new, freely available software for easy determination of strain from 4D-ED data. The authors convincingly explain why their approach is beneficial for TEM users and compare their results with those of an HRTEM instrument. Therefore, a description of the software and of the underlying methods is for sure of major interest for the readers of nanomaterials. Only a few, minor points need to be resolved before the paper is ready for publication.
1. line 5: The address of the first two authors is not complete.
2. line 76: "HAADF" is only defined in line 81
3. line 101: The authors should think about putting the software on a more suitable repository such as GitHub.
4. lines 105 - 130: References for lots of parameters (energy surface density, Poisson ratio for Si, crystallization speed, etc.) are missing. It is not described how, for example, the thickness of the Si:B layer or the ratio of active and total incorporated B atoms was determined.
5. line 112: "chemisorption", not "chemisorbtion"
6. line 153: "with"?
7. Figure 1: The scale bar for (c) is missing. What is the size of the image from which the diffraction pattern in (d) was taken?
8. line 180: Can the authors add a few words about the peek-search algorithm?
9. line 199: Instead of giving the age of the author's laptop, more specific information should be provided, such as memory or CPU or processing power in general.
10. line 210: Fig. 2c is missing an "N" or "L".
11. lines 253-261: The authors should think about shortening this paragraph, which essentially describes that the strain is mapped to a color scale.
12. Figure 4: Even though the paper is about the method and not about the sample, the authors should still add a few lines about the significance of the results obtained (strain distribution in Figs. 4 and 5, depth distribution in Figure 6, etc.).
13. line 290: References for the used values should be added.
Due to these minor shortcomings, I recommend that the paper should be published after a minor revision.
Reviewer 2 Report
|
|
|
|
Comments |
|
|
This work describes a new method to measure strain over a large area of TEM lamella. Although this work will provide an important alternative to measure strain in single crystals, certain things need to be addressed to consider this work for publication in the Nanomaterials.
1. The manuscript lacks clarity over 4D-ED and fails to convince the importance of the study, what exactly it is, and why one should worry about 4D-ED over other available tools. 2. This work is executed using Si:B and the active incorporation of B in Si is about 76%. This amount is so high that it contains a low concentration of Si in Si:B. Although such a high concentration of dopant is needed for superconductivity, it is unlikely that other semiconductors will possess such a high dopant concentration irrespective of the application. Now the question is, will this method work for Si or other semiconductors with low dopant concentration? Low dopant concentration might induce less strain on the crystal and so wonder how this method will work in those cases. I suggest including the results of one such representative example. 3. Since the STEM image has poor resolution, it is interesting to consider the background while collecting the 4D-ED. Do background signals affect the dataset? If yes, how this work addressed this problem? 4. Block diagram presented in Figure 3 can be much clearer and easy to understand. 5. Figures 4a and 4b look like screenshots and can be presented in a better way for a clear understanding. 6. Abstract: Please remove the link present at the end of the abstract. Recommendation: Minor revision |
|

Round 2
Reviewer 1 Report
The authors of "Strain measurement in single crystals by 4D-ED" have addressed all of my previous comments. Only a few minor points remain open:
1. line 59: "... two neighboring pixels) ...", not "... to neighboring pixels) ..."
2. line 75: "existing", not "exaisting"
3. line 136: do the authors mean "cleaving" or "cliving"?
Because of this, I recommend that the paper should be accepted in present form.